# Physiologically Based Pharmacokinetic Modeling of Extracellular Vesicles

**DOI:** 10.3390/biology12091178

**Published:** 2023-08-29

**Authors:** Prashant Kumar, Darshan Mehta, John J. Bissler

**Affiliations:** 1Division of Biochemical Toxicology, National Center for Toxicological Research, United States Food and Drug Administration, Jefferson, AR 72079, USA; darshan.mehta@fda.hhs.gov; 2Department of Pediatrics, Division of Pediatrics Nephrology, University of Tennessee Health Science Center, Memphis, TN 38103, USA; jbissler@uthsc.edu

**Keywords:** extracellular vesicles, exosomes, nanoparticles, PBPK modeling, pharmacokinetics, toxicity, drug delivery

## Abstract

**Simple Summary:**

Extracellular vesicles (EVs) are cell-derived structures that play a vital role in intercellular communication and have potential as drug delivery platforms. Physiologically based pharmacokinetic (PBPK) modeling can be employed to predict the behavior of EVs in the body, including absorption, distribution, metabolism, and excretion. This information is crucial for assessing the quality, safety, and efficacy of EV-based therapeutics. By integrating data on EV characteristics and physiological processes, PBPK models can optimize drug delivery system design, such as EV size and composition, administration route, and drug dosage. Future research may benefit from computer-based modeling approaches in EV-based therapeutic development.

**Abstract:**

Extracellular vesicles (EVs) are lipid membrane bound-cell-derived structures that are a key player in intercellular communication and facilitate numerous cellular functions such as tumor growth, metastasis, immunosuppression, and angiogenesis. They can be used as a drug delivery platform because they can protect drugs from degradation and target specific cells or tissues. With the advancement in the technologies and methods in EV research, EV-therapeutics are one of the fast-growing domains in the human health sector. Therapeutic translation of EVs in clinics requires assessing the quality, safety, and efficacy of the EVs, in which pharmacokinetics is very crucial. We report here the application of physiologically based pharmacokinetic (PBPK) modeling as a principal tool for the prediction of absorption, distribution, metabolism, and excretion of EVs. To create a PBPK model of EVs, researchers would need to gather data on the size, shape, and composition of the EVs, as well as the physiological processes that affect their behavior in the body. The PBPK model would then be used to predict the pharmacokinetics of drugs delivered via EVs, such as the rate at which the drug is absorbed and distributed throughout the body, the rate at which it is metabolized and eliminated, and the maximum concentration of the drug in the body. This information can be used to optimize the design of EV-based drug delivery systems, including the size and composition of the EVs, the route of administration, and the dose of the drug. There has not been any dedicated review article that describes the PBPK modeling of EV. This review provides an overview of the absorption, distribution, metabolism, and excretion (ADME) phenomena of EVs. In addition, we will briefly describe the different computer-based modeling approaches that may help in the future of EV-based therapeutic research.

## 1. Introduction

Extracellular Vesicles (EVs) are nonliving, small, membrane-bound vesicles that are released by cells and play an important role in intercellular communication. They are classified based on their size and biogenesis: Microvesicles (MVs), now also defined as large EVs (l-EVs) [1], are derived from the plasma membrane and range in size from 0.1 to 1 micrometer. Exosomes, now also defined as small EVs (s-EVs) [1] are derived from endosomes and range in size from 30 to 100 nanometers [1]. In addition, EVs bigger than 1 micrometer were released following apoptosis (i.e., Apoptotic Bodies) or by cancer cells (namely oncosomes) [2,3]. EVs are present in all body fluids, including blood, urine, saliva, and cerebrospinal fluid, and carry a diverse range of biomolecules, such as lipids, proteins, and nucleic acids [4]. The biogenesis of EVs is a complex process that varies depending on the cell type and the physiological or pathological context. In general, the biogenesis of EVs involves the formation of a multivesicular body (MVB) within the cell, which then fuses with the plasma membrane, releasing the vesicles into the extracellular space [4]. The biogenesis of EVs is described in detail in later sections.

EVs play a role in a wide range of physiological and pathological processes, including cell growth, differentiation, and death, as well as in the development and progression of various diseases [5,6]. They are also being investigated as potential diagnostic and therapeutic tools [7]. The mechanisms of intercellular communication mediated by EVs are not fully understood, but it is thought that the transfer of biomolecules such as proteins, lipids and nucleic acids between cells plays a critical role [1]. This transfer of biomolecules can alter the gene expression and protein expression of the recipient cell, leading to changes in cell behavior, such as proliferation, differentiation, or apoptosis [1].

In a disease context, EVs were found to be involved in cancer progression and metastasis, as they can transfer oncogenic proteins and genetic material to surrounding cells, thereby promoting the growth and spread of cancer [8]. EVs are also being explored as a means of delivering drugs and genetic material to specific cells and tissues, as they have the ability to target specific cell types and tissues and can protect the payload from degradation by the body’s immune system [9]. Due to their ability to carry a diverse range of biomolecules, EVs are also being investigated as potential diagnostic tools, as they can be used to monitor disease progression and as a source of biomarkers for early detection of diseases [10]. It is important to note that EV research is still in its early stages, and more research is needed to fully understand the mechanisms and potential applications of these vesicles.

To explore the use of EVs as a treatment option, a detailed understanding of their pharmacokinetics is essential. This can be achieved through the application of physiologically based pharmacokinetic (PBPK) modeling, which can provide information on the absorption, distribution, metabolism, and excretion (ADME) of EVs. This review is the first of its kind in that it describes the detailed application of PBPK modeling in EV research. This manuscript is structured in a way that allows for a thorough understanding of both EVs and PBPK modeling. Initially, the review will discuss the details of EVs, including their structure, biogenesis, composition, and related technologies used in EV research. The later part of the manuscript will focus on the ADME properties of EVs and provide a detailed description of PBPK modeling and its application in EV research.

## 2. Extracellular Vesicles: A Brief Sketch

When first discovered, EVs were thought to be simply a mechanism for cells to clear away debris by expelling unwanted materials into the extracellular space [11]. However, in recent years, as research has progressed and technology has advanced, it has become clear that these nanosized particles possess a wide range of properties that were previously undiscovered. These particles, which are released by most cells, contain a variety of molecular signals in the form of proteins, mRNA, miRNA, DNA, and lipids [1,8]. These signals play a crucial role in various cellular functions throughout the human body.

EV research was initially undervalued, as reflected in the limited number of publications from 2003 to 2012. However, as techniques for isolating and characterizing EVs have improved, research on EVs has gained more attention, as evidenced by the large increase (around 10 times) in publications on EVs in the current decade (2013–2022) (Figure 1). This trend is expected to continue to grow exponentially in coming years. PBPK modeling is not a new approach and has been used in literature since 1970, but there is still a lack of literature on the application of PBPK modeling in EV research in the past decade, with only one literature found in the past ten years [12]. The article by Modh et al. (2021) [10] is not a dedicated PBPK study, but rather a search output in PubMed based on specific keywords. It focuses on the in vitro to in vivo correlation within a compartmental model, which is one application of PBPK modeling.

This highlights (Figure 1) the potential for PBPK modeling to greatly enhance EV research. The number of publications on EVs alone is much greater than the number of publications on EVs in combination with PBPK modeling. Three different key terms, i.e., PBPK, Extracellular Vesicles, and Extracellular Vesicles + PBPK, were searched in PubMed to retrieve the number of publications during the mentioned time frame with no filters applied.

As EVs contain informational materials in the form of nucleic acids, proteins, and metabolites produced by host cells, researchers have leveraged this property of EVs to investigate disease biomarkers and therapeutics. Studies have demonstrated the great potential of EVs as a cell therapy in preclinical and clinical settings [13,14,15]. In order to fully harness the safety and efficacy properties of EVs on a large scale, it is important to have a detailed understanding of their pharmacokinetics. PBPK modeling is a tool commonly used by drug developers, which is composed of mathematical equations and can predict the pharmacokinetics of drugs or chemical entities [16]. The application of PBPK modeling to EVs will be discussed in later sections.

## 3. Structure and Composition of EVs

EVs are composed of a lipid bilayer enclosing inner lumen encasing bioactive molecules derived from donor cells [17]. The lipid bilayer of EVs is composed of a variety of lipids, including phospholipids, sphingolipids, and cholesterol (Figure 2). The phospholipid components of the EV membrane are similar to those of the plasma membrane from which they are derived, but the lipid composition of the EV membrane may be different from that of the parent cell [17]. The composition of EVs can vary depending on the cell type and the conditions under which they are produced. For example, exosomes derived from cancer cells may have a different composition compared to exosomes derived from healthy cells [17]. The intraluminal vesicles of EVs contain a variety of biomolecules, including protein, nucleic acid, lipids, and metabolites.

Proteins: EVs contain a wide variety of proteins, including enzymes, signaling molecules, structural proteins, and cytoskeletal proteins. The protein content of EVs can vary depending on the cell type and the conditions under which they are produced [18]. In addition, there are various transmembrane proteins (such as CD9, CD63 and CD81) and various kinds of receptor proteins that are also present on the EV surface [19,20,21]. Nucleic acids: EVs contain small amounts of RNA, including mRNA, miRNA, and long non-coding RNA [22]. The RNA content of EVs can vary depending on the cell type and the conditions under which they are produced. Metabolites: EVs contain small amounts of metabolites, such as sugars and amino acids [22].

The content of EVs can also include various other biomolecules such as signaling molecules. These biomolecules are involved in a wide range of physiological and pathological processes, including cell growth, differentiation, and death, as well as in the development and progression of various diseases [17].

## 4. Biogenesis of EV

The biogenesis of EVs refers to the process by which these vesicles are formed and released from cells. There are two main classes of EVs, MVs and exosomes, which are distinguished by their size, origin, and biogenesis [19]. MVs are formed by the shedding of plasma membrane fragments and can be released from cells through various mechanisms such as exocytosis and ectosomes [19]. They are formed by the outward budding of the plasma membrane, which results in the formation of small vesicles ranging in size from 0.1 to 1 micrometer. Exosomes originate from the endosomal system, emerging as intraluminal vesicles (ILVs) within multivesicular bodies (MVBs) [19].

The biogenesis of EV is a complex process that involves multiple steps and the participation of different components. One of the key players in the biogenesis of exosomes is the endosomal sorting complex required for transport (ESCRT) machinery, which is composed of several other proteins including HSP 90, ALIX, and TSG 101 [19,23]. HSP 90 is a chaperone protein that is responsible for the folding and stability of other proteins, and plays a role in the biogenesis of exosomes by interacting with the ESCRT machinery. As mentioned earlier, the ESCRT complex is a group of proteins that are responsible for sorting and packaging cargos, including proteins and lipids, into vesicles for transport to different cellular locations [24]. The ESCRT machinery is composed of four main subcomplexes: ESCRT-0, ESCRT-I, ESCRT-II, and ESCRT-III [23]. These subcomplexes work together to recognize, recruit and sort cargos into intraluminal vesicles (ILVs) in the endosomal compartment. ESCRT-0 is responsible for recognizing and recruiting the cargos to be sorted, ESCRT-I is responsible for the formation of the ILVs, ESCRT-II is responsible for the scission of the ILVs from the endosome and ESCRT-III is responsible for the final fission of the vesicles from the plasma membrane, releasing the exosomes into the extracellular space [19]. Further, ALIX is involved in the formation of intraluminal vesicles (ILVs) within the endosomal compartment, and plays a role in the biogenesis of exosomes by interacting with the ESCRT-III subcomplex [19]. In addition, TSG 101 is also involved in the final fission of the vesicles from the plasma membrane and plays a role in the biogenesis of exosomes by interacting with the ESCRT-III subcomplex [19]. It is important to note that not all exosomes are formed by the ESCRT-dependent mechanism. Exosomes can also be formed by other mechanisms such as shedding of the plasma membrane and direct fusion between the plasma membrane and endosomes [19].

## 5. Technology Advancement and EV Research

The advancement of technology plays a crucial role in the progress of life science research. As technology improves, so too does our understanding and capabilities in specific areas of life science. Research on EV relies heavily on technological advancements, from the generation of EVs to their isolation and characterization. The development of more advanced techniques for isolating and characterizing EVs has allowed for a greater understanding of their properties and functions. For example, the development of new methods for isolating EVs, such as ultracentrifugation, size-exclusion chromatography, and affinity-based methods, has allowed for the isolation of EVs with greater purity and yield [25]. This has enabled researchers to study the specific biomolecules and signaling pathways that are associated with EVs.

Isolation methods: There are a variety of methods used to isolate EV from biological fluids, such as blood, urine, and cerebrospinal fluid. Each method has its own advantages and limitations, and the choice of isolation method will depend on the specific characteristics of the sample and the research question being investigated. Some of the commonly used isolation methods include:

Ultracentrifugation: This method is still considered as a gold standard for EV isolation. It involves centrifuging a biological sample at high speeds (100,000× *g*) to separate EVs from other components in the sample based on their size and density [20,26]. This method is widely used due to its simplicity, high yield, and purity of the isolated EVs [26]. However, it has some limitations, such as the need for large volumes of starting material and the potential for loss of some EV subpopulations [20]. It also requires highly trained personnel to handle the ultracentrifugation.

Size-exclusion chromatography (SEC): This method separates EVs based on their size by passing a biological sample through a column filled with beads of a specific size [20]. One of the most common SEC available commercially is a qEV column from Izon Biosciences [27]. These columns have beads with a defined pore size that only allows the passage of particles smaller than that pore size [28,29]. This method is widely used due to its high yield and purity of the isolated EVs. However, it has some limitations, such as the need for large volumes of starting material and high cost.

Affinity-based methods: This method uses specific binding molecules, such as antibodies, to selectively isolate EVs from a biological sample [30]. This method is widely used due to its high specificity and selectivity in isolating EVs [30]. However, it has some limitations, such as the need for large volumes of starting material, high cost, and the potential for loss of some EV subpopulations.

PEG precipitation: Precipitation methods are utilized to isolate EVs by exploiting their solubility characteristics. Addition of polyethylene glycol (PEG), with an average molecular weight of 10 kDa, is commonly used to increase hydrophobic interactions among EVs and between EVs and other substances. Consequently, water is excluded, and EVs can be pelleted after incubation and via a single low-speed centrifugation step [31].

Ultrafiltration: Ultrafiltration is a separation method based on molecular size and is one of the simplest methods for exosome separation. Exosomes are obtained by removing impurities through one or more filtering membranes with different pore sizes or the molecular weight cut off (MWCO). The pollutants larger than MWCO are quantitatively held back by the filtering membrane, while other components (exosomes) smaller than the MWCO can pass through the filtering membrane structure along with the permeate. Depending on the driving force, ultrafiltration can be classified as electric charge, centrifugation, and pressure [32].

Characterization methods: Additionally, the advancement of imaging technologies, such as transmission electron microscopy (TEM) and scanning electron microscopy (SEM), has allowed for the visualization of EVs at higher resolutions, providing insights into their structure and biogenesis [33]. The advancements in the field of genomics and proteomics have also played a critical role in EV research, by allowing the characterization of the molecular composition of EVs at a large scale [18,34]. This has led to the identification of new biomarkers and therapeutic targets for various diseases. In addition to the imaging technology development, biophysical methods also play an important role in the characterization of EV. Some of them are Western blot analysis, zeta sizer, nanoparticle tracking analysis (NTA), Tunable Resistive Pulse Sensing (TRPS), fluorescence-activated cell sorting (FACS), and Dynamic Light Scattering (DLS). The major methods for isolating and characterizing EVs are illustrated in Figure 3.

Furthermore, new technologies were developed in the field of nanotechnology, which allows for the manipulation of EVs for therapeutic applications, such as targeted drug delivery, by modifying the surface of EVs [35]. In recent years, advancements in modeling software and the integration of machine learning and artificial intelligence have greatly impacted the field of drug discovery research by revolutionizing the way it is conducted, especially in disease therapeutics [36,37].

### 5.1. Extracellular Vesicles and Liposomes—Similarities and Differences

The interest in lipid-based drug delivery systems has grown in recent years, with researchers exploring new ways to improve drug delivery. Although the use of EVs as a drug delivery system is still an emerging field, liposomes are widely studied and well-established as a drug delivery system [38,39]. Given the similarities in physiochemical properties between EVs and liposomes, studying liposomes can provide valuable insights into the potential applications of EVs in drug delivery [40]. In this section, we will compare the similarities and differences between liposomes and EVs (Figure 4), highlighting the key takeaways from each field that are relevant to pharmacokinetics [40].

EV and liposomes are both spherical, bilayer structures composed of lipids, but there are some important differences between them (Figure 4).

Origin: EVs are naturally occurring structures that are released by cells into the extracellular space [41], while liposomes are artificially created in the laboratory [42]. Composition: EVs are composed of lipids, proteins, and nucleic acids that are derived from the host cells, while liposomes are composed of synthetic lipids that are not derived from host cells [40]. Biogenesis: EVs are formed via the endosomal-sorting complex required for transport (ESCRT) machinery [23], while liposomes are formed via the process of hydration and sonication of lipids [42]. Function: EVs are shown to play a role in intercellular communication [1], while liposomes are primarily used as drug delivery vehicles [39].

In our context we would be more interested in the similarities in the pharmacokinetics between Liposome and EVs. Some of the similarities include: (1) Size: Both EVs and liposomes have similar sizes, ranging from 20 to 1000 nanometers. This small size allows them to evade detection by the immune system and to penetrate deep tissues [40]. (2) Surface markers: Both EVs and liposomes can be engineered to express surface markers that allow for their targeting to specific cell types or tissues [40]. (3) Long circulation time: Both EVs and liposomes have a long circulation time in the bloodstream, allowing them to reach distant sites in the body. (4) Biocompatibility: Both EVs and liposomes are biocompatible, meaning that they do not cause adverse reactions in the body [40]. (5) Drug encapsulation: Both EVs and liposomes can encapsulate drugs, protecting them from degradation and increasing their circulation time [38,43]. (6) Both can be used as a vehicle for cell-free therapeutics and gene therapies. (7) Once taken up the cells, they both follow the endocytic pathway and release their payload [40].

Currently there are no review articles that specifically focus on the pharmacokinetics modeling of EVs. However, liposomes are extensively studied in this regard, and their similarities in structure and the endocytic pathway make them a valuable resource for understanding the pharmacokinetics of EVs. In this manuscript, we will use the knowledge gained from studying liposomes to better understand the potential applications of EVs in drug delivery.

### 5.2. Extracellular Vesicles and Nanoparticles: PBPK Modeling

Our approach involves leveraging the insights and modeling strategies obtained from prior investigations of nanoparticles and pharmacokinetic modeling to ascertain their potential transferability to the realm of EVs. Dr. Lin has performed a pioneer study by developing a PBPK model of gold nanoparticles in rats [44]. He further compared the model development approach between the traditional and new route specific data. He discovered that the traditional approach of PBPK modeling for small molecules cannot be compared with nanoparticles due to different physiochemical characteristics and metabolic behavior [44]. Hence, multi-route PBPK models for nanoparticles should be developed using route specific data. For PBPK parameterization and optimization, Dr. Lin utilized a Bayesian approach with Markov chain Monte Carlo (MCMC) simulation for the nanoparticles, and later converted to a web-based interface using the Shiny package of R [44]. He further employed quantitative structure–activity relationships (QSAR) and implemented multivariate linear regression models to establish robust predictions for key biodistribution parameters pertaining to specific routes of administration [44]. The results demonstrated that, irrespective of the administration route, the primary determinants governing endocytic/phagocytic uptake rates were identified as the size and surface area of the gold nanoparticles. Furthermore, the Zeta potential emerged as a significant parameter in accurately estimating the exocytic release rates after intravenous (IV) administration [44]. This study can provide a foundation to develop a PBPK for EVs.

## 6. EVs in Clinical Research

Recent advancements in technology for EV isolation and characterization have led to an increase in the use of EVs in clinical trials [45]. EVs were shown to have potential as a therapy for various diseases, with studies showing the use of mesenchymal stem cell (MSC)-derived EVs in regenerative medicine are a suitable model for experimental and clinical trials [46]. In clinical trials, the application of EVs is typically categorized into five different areas, with approximately 50% of trials focusing on the use of EVs as biomarkers, and the remaining 50% investigating the use of EVs in therapy, cancer vaccines, drug delivery, and analysis [47]. Here, we have listed some important clinical trials related to EVs that are either completed or ongoing, divided into three different disease conditions: cancer, coronavirus, and cardiovascular diseases, as shown in Appendix A.

## 7. ADME of EVs

The ADME processes are critical in determining the disposition of a pharmaceutical entity within an organism. Despite the growing interest in EVs as a potential drug delivery system, the metabolism of EVs has not been extensively studied. In this review, we will consider EVs (loaded with the active pharmaceutical compound) as a single entity and will discuss the absorption, distribution, and excretion parameters.

### 7.1. Absorption

The most important principle in pharmacokinetics theory is drug absorption, which refers to the transportation of unmetabolized drugs from the site of administration to the body’s circulatory system [48]. Absorption is a crucial parameter in pharmacokinetics, as it determines how a drug or pharmaceutical formulation moves from the site of administration to its site of action. Although research on the pharmacokinetics of EVs is limited, recent studies have provided insights into how EVs are absorbed by cells and tissues [49]. Upon contact with target cell membranes, EVs transfer signals that regulate various biological events [50]. In this process, cell membrane-mediated transport plays a crucial role in the transport of EVs and their contents to target cells [51]. EVs are released from cells into surrounding body fluids, and their absorption or uptake is mediated by multiple factors, which can be either receptor-dependent or receptor-independent. It is possible that EVs can be internalized by more than one mechanism in the same cells [52]. The absorption or internalization of EVs at the cellular level takes place through various mechanisms, which are detailed below.

#### 7.1.1. EV Absorption by Fusion

EV absorption by fusion is a widely studied and simple mechanism in which the EV directly fuses with the target cell membrane [53]. This process, known as endocytosis-independent uptake, is mediated by the presence of specific lipids or proteins on the EV surface that are complementary to those found on the target cell membrane [53]. This mechanism of EV uptake is relatively fast and efficient, allowing for the rapid transfer of material from the EV to the target cell. Studies have shown that this mechanism of EV absorption is mediated by the presence of specific lipids such as phosphatidylserine and sphingomyelin on the EV surface, as well as by the presence of specific proteins such as tetraspanins and integrins [54,55]. It is important to note that the mechanism of EV absorption by fusion is still under investigation, and more research is needed to fully understand the mechanisms that govern this process and its applications in different fields such as drug delivery. However, studies have shown that this process is dependent on lipid rafts and caveolar endocytosis, which can be inhibited using filipin III in human endothelial cells [56]. Additionally, various proteins present on the outer surface of EVs, such as tetraspanin, have been shown to play a role in the fusion process [57]. Tetraspanin are not only present on the surface of EVs as markers, but also mediate various cellular functions, such as T cell activation and fertilization in humans, specifically during the fusion of oocyte and sperm [58].

#### 7.1.2. Phagocytosis

Phagocytosis is a process by which cells, particularly immune cells such as macrophages and dendritic cells, engulf and internalize larger particles, including pathogens and cellular debris. This process can also occur for EVs, in which the EV attaches to the cell membrane and is then engulfed by the cell through a process that involves the formation of a phagosome [59]. This process is mediated by receptors on the cell surface, such as scavenger receptors, integrins, and tetraspanins, which bind to specific molecules on the EV surface [60]. Phagocytosis of EVs is facilitated by PI3K and actin protein, which are involved in the formation of the phagosome and the internalization of the EV [61]. This process is important for the immune response and can also be used by other cell types such as cancer cells. It is important to note that, Phagocytosis is a dynamic process, which can vary depending on the type of cell and the specific conditions of the environment [62]. For example, the rate of phagocytosis can be affected by the presence of other particles, by the presence of certain signaling molecules, and by changes in the cell’s environment.

#### 7.1.3. Clathrin-Mediated Endocytosis

Clathrin-mediated endocytosis (CME) is a mechanism by which cells internalize extracellular material, including EVs, through the formation of clathrin-coated pits on the plasma membrane [61]. These coated pits are composed of a protein coat made up of clathrin and other associated proteins, which surrounds the material to be internalized [63]. The process is mediated by receptors on the cell surface that bind to specific molecules on the EV surface, such as tetraspanin, and then recruit clathrin to the site of internalization [61]. The clathrin coat then forms a pit that buds off from the plasma membrane and brings the EV into the cell. Inside the cell, the clathrin coat is removed and the EV is released into the cytosol or directed to lysosomes for degradation [61]. This process is important for the internalization of EVs and their cargo into the cell and is also involved in the internalization of other membrane-bound vesicles and particles [61]. This mode of EV internalization is mostly seen in tumor cells, cardiomyocytes, macrophages, and neural cells [61,64].

#### 7.1.4. Caveolin-Mediated Endocytosis

Caveolin-mediated endocytosis is another mechanism by which EVs can be internalized by cells. This mode of internalization is mediated by caveolin proteins, which are integral membrane proteins found in the plasma membrane of many cell types [65]. Caveolae are small cave like structures (caveolae) formed by the invaginations of the plasma membrane, which are rich in cholesterol and sphingolipids and provide the necessary environment for caveolin-mediated endocytosis to occur [65]. Caveolae functions are specialized platforms for endocytosis, where the caveolin proteins interact with specific receptors on the surface of the EVs [61]. This interaction leads to the formation of a caveolar vesicle, which is then pinched off from the plasma membrane and brought into the cell. Once inside the cell, the caveolar vesicle is targeted to the endosome for further processing or degradation [61]. This mode of internalization is commonly seen in endothelial cells [66], smooth muscle cells [67], and fibroblasts [68]. Research has shown that deletion of CAV1 gene causes the reduction in EV uptake in epithelial cells [69].

#### 7.1.5. Lipid-Raft Mediated

As the name suggests, this mode of EV transport requires the involvement of a lipid raft. Lipid rafts are the complex assembly of proteins and lipids that float within the plasma membrane and are mostly involved in signal transduction [70]. Lipid rafts mainly consist of cholesterol, sphingolipid, and glycosylphosphatidylinositol (GPI)-anchored proteins [70]. EV uptakes are greatly reduced when a dendritic cell is pre-exposed to fumonisin B1 and N-butyldeoxynojirimycin hydrochloride, which are known to reduce the sphingolipid part in the lipid raft in the plasma membrane [51]. This highlights the crucial role of the lipid microdomain of the exosomal membrane and its importance in EV-related regulatory mechanisms [71].

#### 7.1.6. Macropinocytosis

Macropinocytosis is a form of endocytosis characterized by the formation of ruffles and cups in the plasma membrane [72]. During this process, the plasma membrane invaginates, creating small vesicles that are internalized and enter the endocytic pathway. Unlike phagocytosis, direct contact with the internalized material is not required. Macropinocytosis is mediated by the proteins rac1, cholesterol and actin. Further, rac1 activity is dependent on changes in osmotic pressure resulting from ion fluxes (Na^+^/H^+^/Ca^2+^) [73]. Studies have found that macropinocytosis is a major mechanism for the transfer of EVs from oligodendrocytes to microglia. Inhibiting the Na^+^/H^+^ ion flux was shown to decrease EV uptake by microglial cells [74]. Additionally, the use of pharmacological inhibitors, such as NSC23766 (an inhibitor of Rac1) and amiloride (which blocks Na^+^/H^+^ ion flux), was shown to reduce EV uptake in microglial cells, further suggesting a role for macropinocytosis in EV uptake [74].

The absorption of EVs in the systemic circulation is highly dependent on the route of administration. We will briefly discuss oral and IV routes of EV administration, as they are the most common.

#### 7.1.7. Oral Exposure

Oral administration is the most preferable mode of drug administration in humans for several reasons. Some of the most common reasons are that it does not directly damage the skin, is not painful for prolonged therapy, can be self-administered, does not require special skills to administer, is convent, and economical [75,76]. Absorption of EVs after oral exposure is an area of active research. The oral route is one of the simplest and least invasive methods for administering drugs and therapies, but it also presents challenges for EV-based therapies. One factor that affects the oral absorption of EVs is the size of the vesicles. Oral delivery of sodium glycocholate-liposome showed better protection of insulin against proteases, leading to improved bioavailability of insulin [77,78]. These outcomes were dependent on the size of the particles [78]. As EV and liposomes have comparable size and surface characteristics, similar behaviors could be expected when EVs are administered orally. Smaller EVs have a higher chance of being absorbed through the gut wall, while larger EVs may be trapped in the gut lumen and eliminated through feces. Another factor that affects the oral absorption of EVs is the composition of the vesicles. The lipid bilayer of EVs can protect the contents from degradation, but it can also reduce EVs ability to penetrate the gut wall. The presence of proteases in the gut can also degrade the EVs, reducing their effectiveness [78]. Finally, the efficiency of oral EV absorption can be influenced by the gut microbiome, which can affect the composition and function of the gut epithelium [79]. Despite these challenges, there is still a great deal of interest in developing EV-based therapies that can be delivered orally, as this would provide a safe, simple, and convenient way to deliver therapeutic agents to the patient. For example, oral administration of bovine milk-derived EV could regulate the proteomics profile in the liver tissues in mice [80]. Further, the same EV could reduce the colorectal and breast tumor in mice [80]. Research in this area is ongoing, and it is hoped that the development of new technologies and methods will enable the successful oral administration of EV-based therapies in the future.

#### 7.1.8. Intravenous Injections

IV administration of EV is an effective and efficient way to deliver therapeutic agents to the body, as the drugs are directly introduced into the bloodstream [81]. In the case of IV exposure of EVs, the entire dose of the therapeutic agents is rapidly absorbed into the systemic circulation, allowing for quick and efficient distribution to target tissues. When injected intravenously, EVs can cross the blood–brain barrier and successfully deliver the cargo protein to the target site, which could serve as a potential therapeutic approach for CNS disorders [82]. Exosomes can also serve as an attractive therapeutic tool for traumatic brain injury by reducing inflammation when infused intravenously [83].

#### 7.1.9. Other Routes

In addition of oral and IV, subcutaneous and intraperitoneal administration of EV also leads to the fast systemic clearance and accumulation in the liver, lungs, spleen, and gastrointestinal tract [84,85]. In addition, the inhalation route of EV exposure shows therapeutic promise for pulmonary fibrosis and COVID-19 patients [86,87].

### 7.2. Distribution

The distribution of EVs in the human body depends on various factors, such as the size, charge, and surface properties of the EVs, as well as the presence of specific receptors on target cells. For example, when the surface protein of exosomes was modified using near infrared fluorophore, the targeting properties and its biodistribution was improved with less nonspecific uptake [88]. Further this surface modification also leads to the rapid renal clearance and altered pharmacokinetics profile [88]. Once EVs enter systemic circulation, they can be transported to various organs and tissues throughout the body. S-EVs are able to pass through the endothelial barrier and enter the circulatory system, allowing for distribution to various organs and tissues [89,90]. L-EVs may be restricted to the microvasculature of specific organs and tissues [91]. In the bloodstream, EVs are known to interact with cells, such as platelets, monocytes, and red blood cells, and can accumulate in organs such as the liver, spleen, and lymph nodes [92]. In addition, EVs were found to accumulate in certain disease states, such as cancer, where they may play a role in the progression and spread of the disease. It is also known that the distribution of EVs is influenced by the presence of specific receptors on target cells, which can aid in their targeting to specific organs and tissues [88].

While it is still not well understood about the distribution dynamics of EVs in the human body, a combination of various techniques such as flow cytometry, nanoparticle tracking analysis, and imaging can be used to determine the distribution of EVs in the body. In the current context, PBPK modeling can be applied to predict the distribution of EVs in the human body after administration. This includes factors such as the biodistribution of EVs, their clearance from the body, and the pharmacokinetics of the informative materials present within the EVs. PBPK modeling can also be used to predict the pharmacokinetics of EVs in different patient populations and to evaluate the safety and efficacy of EV-based therapies.

### 7.3. Clearance

Clearance is a combined term used for metabolism and excretion in this case, as there may be no metabolism involved for exosome clearance. In the case of EVs, metabolism refers to the changes in the physicochemical properties of the EV [93]. EVs are generally metabolized via the endocytic pathway, which is the process by which cells internalize molecules from the extracellular environment. The endocytic pathway can be divided into several stages, including endocytosis, transport, and degradation. During endocytosis, the EVs are taken up by the cell through mechanisms such as clathrin-mediated endocytosis or caveolae-mediated endocytosis [94]. After endocytosis, the EV is transported to different intracellular compartments, such as endosomes or lysosomes, where it is degraded [94]. The changes in the physical behavior of EVs, such as the dispersal of the cargo they are carrying, can affect the ADME of the EV. For example, the size and shape of the EV, as well as the composition of its cargo, can affect its ability to cross the cell membrane and enter the cell. Additionally, the rate of endocytosis and the efficiency of the endocytic pathway can also affect the metabolism of the EV.

Research conducted in mice found that when melanoma derived EVs were injected intravenously, the clearance of EVs from the systemic circulation was rapid with macrophages from the liver, kidneys, and spleen being associated in the clearance [61]. This indicates that macrophages from these organs play a critical role in the removal of EVs from circulation [95]. Additionally, another study found that the mode of EV exposure, IV or intratumoral, affects the EV’s clearance [61]. When EVs were delivered by IV injection, the clearance of EVs from the systemic circulation was rapid. In contrast, when EVs were delivered by intratumoral injection, EVs stayed in the tumors longer, indicating the different routes of administration of EVs can affect the EV’s clearance [61]. Overall, research showed that macrophages play a crucial role in the clearance of EVs from the systemic circulation. This highlights the importance of understanding the clearance mechanisms of EVs, as it can have implications for the design of EV-based therapies and the interpretation of EV-based diagnostics.

Another in vivo study revealed that exosomes, derived from B cells, have a plasma half-life of around two minutes when administered IV [57]. This rapid clearance is likely due to the action of macrophages and other immune cells, which are known to rapidly clear EVs from the bloodstream. However, the study also found that even though the exosomes were quickly cleared from the blood, they were still present in their reservoir organ, the spleen, for two hours [57]. This suggests that exosomes may be taken up and retained by certain organs or tissues, where they can continue to exert their effects. The spleen is known to be a reservoir organ for exosomes, and it is also known that exosomes can affect immune response. Similarly, in another study, it was found that after intranasal administration, the exosomes were found in brain and intestine after three hours [57]. This indicates that exosomes can cross the blood–brain barrier and the blood-intestinal barrier and reach the brain and the gut. These findings highlight the complexity of the dynamics of EV’s distribution and clearance in the body [57]. The rapid clearance of exosomes from the bloodstream, combined with their ability to be taken up and retained by specific organs, suggest that the effects of exosomes may be both short-lived and long-lasting, depending on the route of administration, the organ targeted and the nature of the exosomes. This information is critical for the design of exosome-based therapies and for interpreting exosome-based diagnostics. Like in an earlier study, we have calculated the half-life of EVs using an in vitro system, which uses a mathematical model [29]. We did the uptake study of fluorescent-labelled EV in a cell culture system and found the improved half-life of EV derived from the Pkd1 (92.4 h) mutant cell as compared to the Pkd2 (17.3 h) mutant cell [29]. Similar patterns were also observed in the Tsc2 (13.86 h)-derived EV and Tsc1 (4.62 h)-derived EV.

## 8. PBPK Modeling in EV Research

PBPK modeling is a mathematical and computational approach used to predict the movement of drugs and chemicals within the body [16]. PBPK modeling is a type of compartmental pharmacokinetic model that is considered one of the most realistic models among all the compartmental models [96,97]. It is based on the physiological and biochemical properties of the body, such as blood flow, organ size, and enzyme activity and uses mathematical equations to simulate the movement of drugs and chemicals within the body [16].

PBPK models consist of various compartments, which are indicative of different tissues or organs, interconnected by a central compartment often referred to as blood [98]. These compartments are linked by various physiological and biochemical parameters, such as organ volume, partition coefficient, blood flow, and clearance, to predict the drug’s tissue and blood concentration over time [98]. PBPK models are particularly useful for predicting the ADME parameters from one species to another, making it a powerful tool for drug development and toxicology studies [98,99].

PBPK modeling has been used extensively in the field of drug development to predict the PK of drugs and to guide the design of clinical trials [100]. However, in recent years, PBPK modeling was also applied to the study of EV [101]. This is because EVs have similar ADME processes as drugs, and PBPK models can be used to predict the PK of EVs and to understand the factors that influence their distribution and clearance in the body [93]. PBPK modeling has been used to study the PK of a wide range of drugs and chemicals, and the use of PBPK modeling has expanded to various areas like environmental health, toxicology, and pharmacology [102]. PBPK modeling could be used to study the PK of EVs, specifically exosomes, by considering the size, shape, and composition of the exosomes and the physiological properties of the organs and tissues where they are distributed. These models can help to predict the distribution and clearance of exosomes in the body and to understand the factors that influence their PK.

As EVs and liposomes share many characteristics, there are some studies related to PBPK modeling and liposome-mediated drug delivery. This research shows that PBPK modeling can be useful in understanding the distribution and clearance of EVs in the body and can help to guide the design of EV-based therapies and diagnostic approaches [103].

### 8.1. Type of Data Needed to Make a Successful PBPK Model

The application of PBPK modeling to predict the disposition of EVs is a relatively novel pursuit, and current methodologies for leveraging the available data are largely opportunistic. The fundamental objective of employing such modeling techniques is to extract profound insights from existing experimental data, facilitating the comprehensive characterization and quantification of the various processes governing EV disposition within the body.

In our previous publication [29], we exemplified the utility of PBPK modeling by utilizing data from an in vitro system to construct a mathematical model. This model aimed to elucidate the influence of individual gene deletions on EV production and trafficking dynamics. By conceptualizing a dynamic pool of EVs amenable to uptake, binding, and clearance across wild type and mutant cell lines, we mathematically represented the fluorescence time course data obtained. This modeling endeavor enabled us to derive quantitative parameter estimates governing processes such as uptake, binding, and clearance. These estimations, in turn, facilitated the calculation of EV half-lives in distinct cell lines. This case underscores how fluorescence time course data, conventionally providing qualitative insights, can be harnessed via mathematical modeling to yield quantitative understanding of EV disposition.

Similar strategies can be extended to in vivo contexts, where fluorescence intensity data is gathered from diverse tissues and organs. Furthermore, invaluable datasets, such as EV permeability across various membranes, EV partitioning within organs, and the influence of EV size and morphology on disposition, can bolster the successful application of PBPK modeling techniques in EV research. As the field advances, it is anticipated that both conventional and innovative methodologies will contribute to the acquisition of such data. PBPK modeling is poised to serve as a foundational framework for amalgamating diverse data types, culminating in meaningful predictions regarding EV pharmacokinetics.

### 8.2. PBPK Model Validation Criteria

It should be noted that there can never be a true model for any biological system and that all models involve a certain degree of reasonable approximation or simplification, as achieving an exact representation is inherently challenging. Consequently, it is not uncommon within the realm of modeling that multiple models can equally well elucidate a given system or phenomenon. In such instances, the principle of parsimony comes into play, asserting that unnecessary complexities should be avoided, and preference should be provided to models featuring fewer assumptions and parameters. Models, therefore, are tailored to their intended purposes and are assessed based on their specific objectives.

In the assessment of a physiologically based pharmacokinetic (PBPK) model, several metrics for goodness-of-fit are commonly employed. This includes generating plots that juxtapose predicted outcomes against observed data and calculating statistical measures like the coefficient of determination (R^2^) through linear regression analysis, as well as the mean average prediction error (MAPE).
$$MAPE\;\left(\%\right)=\frac{1}{N}\overset{N}{\underset{i=1}{\sum }}\frac{\left|C_{obs,i}-C_{pred,i}\right|}{C_{obs,i}}\times 100$$

For any given PBPK model, a MAPE value below 10% is indicative of an outstanding prediction. If the MAPE falls between 10% and 20%, the prediction is considered favorable, while exceeding 50% is generally deemed unacceptable. Hence, statistical parameters, such as MAPE, serve as valuable tools for the evaluation of the PBPK model’s performance within the domain of extracellular vesicle EV research.

## 9. PBPK Modeling and Simulation of Extracellular Vesicles Mediated Drug Delivery

EVs are a naturally released container that can be utilized as a drug delivery system. As they are produced in the human body, they could reduce the opsonization. Generally, pharmacokinetic models, such as a PBPK model, are built on three approaches; (i) ‘top down’ approach (ii) ‘bottom up’ approach and (iii) ‘middle out’ approach [104]. ‘Top down’ approach is based on observed experimental data, mainly empirical, with the scope of utilization narrowed down to the range of the input data. ‘Bottom up’ approach is based on our broader understanding of the human body and its mechanisms; it can utilize the in vitro data as input data. [104,105]. ‘Middle out’ approach combines the ‘bottom up’ and ‘top down’ approaches [106]; it allows the utilization of available in vivo data to calculate unknown or uncertain parameters. In this way, the parameters with unknown values are optimized by fitting the unknown model parameters against the experimental data, as was performed in PBPK models for some of the nanoparticles [107]. The PBPK modeling framework provides a structured method for extending findings from experiments involving drug-loaded EVs in mice to anticipate and steer human pharmacokinetics (Figure 5). This contributes to the translation of early-stage research into practical clinical implementations.

We have postulated a general whole body PBPK model based on a bottom-up approach and mechanism-based model for tumor reported in a paper by He et al. [103]. This model structure has the potential to elucidate the disposition of EV and its cargo in biological systems. Our hypothesized PBPK model could be used to investigate the effects of drug related and physiological factors on the disposition of EVs in biological systems. As of now, PBPK models are very commonly used in the small molecule drug development.

### 9.1. Whole Body PBPK Model

In the context of EVs, the goal of a whole-body PBPK model is to simulate the fate of EVs and the biological effects they produce after administration. The model incorporates various physiological and biochemical parameters of the body, including blood flow, organ volume, and the activity of metabolic enzymes. The model is based on the principles of mass balance, which states that the amount of substance in a particular compartment at any given time is equal to the amount of substance that has entered that compartment, minus the amount that exited [109]. The whole-body PBPK model for EVs would include compartments for each body tissue and organ, as well as a compartment for the target of concern such as tumors [110]. The model would also account for the size and number of EVs, as well as their rate of uptake and elimination from each compartment. This model can be used to predict the pharmacokinetics of EVs in various species and to compare the pharmacokinetics between different administration routes (e.g., IV vs. oral), and study the impact of various factors, such as age, gender, and disease state, on the pharmacokinetics of EVs [109].

### 9.2. Simplified PBPK Model

This is also commonly referred to as a minimal PBPK model. It includes only the essential compartments that govern absorption, metabolism, and excretion processes (gut, liver, and kidneys) [111]. A simplified PBPK model of EVs could be a useful tool for understanding the pharmacokinetics of these nanosized biological carriers in various biological systems. A simplified PBPK model is divided into compartments which represent the various physiological systems in the body, such as the blood, liver, and kidneys [112]. These compartments are interconnected by pathways that describe the exchange of EVs between different compartments [111]. The inputs to the model include the dose of EVs, the route of administration, and the size and composition of the EVs. The model then predicts the uptake, distribution, and elimination of EVs in each compartment. The predictions can be compared to experimental data to validate the model and to provide insights into the pharmacokinetics of EVs. The predictions can also be used to predict the pharmacological effects of EVs and to evaluate the safety and efficacy of new EV-based therapies. The principle of mass balance holds true for the simplified (minimal) PBPK model as well.

## 10. PBPK Modeling Software

In recent years, the use of EVs in therapeutics has gained significant interest, and as a result, pharmacokinetic (PK) modeling has become an important tool for understanding the distribution and clearance of EVs in the body. As the number of publications using PBPK modeling for EVs has increased, the availability of different modeling platforms has also grown. The different software platforms for PBPK modeling are available either as open source or closed source. These software platforms are mainly designed for different applications based on the user’s needs and the complexity of the pharmacokinetic analysis. For example, some platforms are more suitable for non-compartmental analysis, while others are meant for complex or multi-compartmental analysis. Here are some of the most widely used PBPK modeling platforms in the pharmaceutical industry and academia. In addition, we have also included a brief comparison between different modeling software in Appendix A.

### 10.1. GastroPlus

GastroPlus is a PBPK modeling software that is widely used in the pharmaceutical industry to predict the pharmacokinetics (PK) of drugs in various species, including humans, rats, dogs, and monkeys [113,114]. It can also simulate the PK of drugs after various routes of administration, such as oral, intravenous, transdermal, and inhalation [114]. GastroPlus is a closed source software and it’s developed by SimulationsPlus company. GastroPlus is highly recommended for the Insilco assessment for gastrointestinal absorption model [115]. In other words, GastroPlus is particularly useful for simulating the PK of drugs that are metabolized by the liver and gut, as it uses detailed models of these organs to predict the drug’s PK [116]. It can also consider factors such as food effects, inter-individual variability, and drug–drug interactions [117].

In addition to its use in drug development, GastroPlus can also be used to model the PK of liposomes and other drug delivery systems, including EV. The software can take into account the size, shape, and composition of the EVs, as well as the physiological properties of the organs and tissues where they are distributed. This allows for the simulation of the distribution and clearance of EVs in the body, and the prediction of the factors that influence their PK. GastroPlus is a user-friendly software that allows for the creation of customized PBPK models and the simulation of various scenarios. The software also has a graphical user interface that allows for the visualization of the results and the comparison of different scenarios. GastroPlus was also used in several studies to model the PK of exosomes.

Although no studies have specifically used GastroPlus to predict the ADME of EVs, a recent study has used GastroPlus in combination with in silico prediction analysis to model the pharmacokinetics of liposome-mediated drug delivery [118]. Given the similarities between liposomes and EVs in terms of their physical properties, it is possible to utilize GastroPlus for EV pharmacokinetic modeling. For example, a study by Hussain et al. investigated the transdermal delivery of rifampicin via elastic liposomes in rats and performed a parameter sensitivity assessment (PSA) using GastroPlus. The PSA helped the researchers to investigate important parameters such as drug dissolution, absorption, and availability to the portal vein [118]. These factors can be influenced by other physicochemical properties of rifampicin such as shape, size, or density of particles. The PSA analysis revealed that drug dissolution did not depend on the particles’ physical properties. They also performed a comparative PK analysis of rifampicin in gel form or oral suspension using GastroPlus simulation software [118].

### 10.2. Simcyp

The Simcyp Simulator is a widely used PBPK modeling platform in the pharmaceutical industry and academia, particularly for small drug molecules, generic drugs, new formulations, and biologics [119]. It is preloaded with various population libraries to meet the different needs of users. The Simcyp Simulator is particularly useful for creating multicompartmental PBPK models, which can accurately describe the ADME of drugs in different body tissues [97,120]. This allows for the simulation of the distribution and clearance of drugs in the body, and the prediction of the factors that influence their PK.

EVs are found in various bodily fluids such as urine, blood plasma, saliva, and milk. They have great potential to be used as novel biomarkers in liquid biopsy. These EVs are often rich in protein and RNA, which reflect the characteristics of their parent cells. One recent study has demonstrated the applicability of the Simcyp Simulator to use EVs as a liquid biopsy. The study found that EVs shed from the liver into human plasma can be used as a pharmacological test to monitor the expression levels of different enzymes and transporters in the liver [121]. The study used the Simcyp Simulator for in silico trials to select the appropriate dose using EV-based liquid biopsy. The authors first utilized multi-omic data to create a link between plasma exosomes and liver tissue expression [121]. They then performed drug trials using Simcyp to simulate the impact of the liquid biopsy input on dose selection. Three CYP3A substrates were evaluated: alprazolam (low hepatic clearance), midazolam (medium clearance), and ibrutinib (high clearance) [121]. The compound files were selected from the Simcyp library. This research shows the potential of using the Simcyp Simulator for the prediction of ADME of EV-based drugs using liquid biopsy. It is important to note that more research is needed to fully understand the applicability of the Simcyp Simulator in EV research; however, this study suggests that it may be useful in predicting the ADME of EV-based drugs and in liquid biopsy applications.

### 10.3. PKSIM

PK-Sim^®^ is another impressive tool for the multicompartmental whole body PBPK modeling [122]. This is a freely available software with a user-friendly interface and is loaded with all relevant parameter values related to common animal models and humans [122]. PK-Sim^®^ has different strategies to build a model; it uses the block method to develop a model. This platform is highly used in academia, industry, and regulatory agencies. PK-Sim^®^ is also compatible with users lacking modeling experience. PK-Sim^®^ is also compatible with the expert modeling tool MoBi^®^, which allows users to perform detailed simulations. PK-Sim^®^ has shown its applicability in several preclinical animal models and various kinds of population such as pediatrics [123] and drug–drug interaction [124].

### 10.4. Berkeley Madonna

Berkeley Madonna is a relatively less expensive mathematical modeling software that provides an easy and intuitive method to model systems of differential equations. It has the potential to be used in various areas of research, including pharmacology and EV research [125]. As of now, there have not been any published studies in PubMed that have specifically used Berkeley Madonna for EV research. However, the software’s capability for modeling dynamic systems, including pharmacokinetics and pharmacodynamics, makes it a potentially valuable tool for EV research in the future [126].

## 11. ADME Mathematical Equations

In our previous study, we found that a mutation in Pkd1 or Tsc2 genes can alter the EV production and trafficking in polycystic kidney disease [29]. We have also described the ADME of EVs and the application of mathematical modeling in EV distribution [29]. In brief, we found that despite having a similar level of binding for the EVs derived from Pkd1 or Pkd2 mutant cell lines, Pkd1-EVs were up-taken 14 times faster and cleared five times slower than Pkd2-EVs. This suggests the involvement of the polycystin-1 gene in the rapid uptake and prolonged half-life. Similarly, EVs derived from Tsc2 mutant cells were taken up at a two times higher rate than the EVs derived from Tsc1 mutant cells [29]. We have depicted the interaction between EVs and renal cells in Figure 6B. We posit here that an EV depot facilitates the binding phenomenon, and the clearance of EVs is mediated by lysosomes. We have also listed out mathematical equations for better understanding, which follows the first order uptake and clearance of EVs [29]. Below is the example of mathematical equations to describe the uptake and distribution of EVs in the blood and in various tissues for a multicompartmental PBPK model.
(1)$$V_{i}\times \frac{dC_{i}}{dt}=Q_{i}\times \left(C_{a}-C_{i}/R_{i}\right)$$For all tissue except lung: Here *C_i_, V_i_, Q_i_* and *R_i_* are the EVs concentration, volume, and partition coefficient of the *i*th tissue respectively. *C_a_* is the EVs concentration in the arterial blood. The partition coefficient, *R_i_* will be determined by calibrating the model using the fluorescence intensity data.
(2)$$V_{lung}\times \frac{dC_{lung}}{dt}=Q_{B}\times \left(C_{v}-C_{lung}/R_{lung}\right)$$For lung tissue: *C_lung_, V_lung_,* and *R_lung_* are the EVs concentration, volume, and partition coefficient respectively of the lung. *Q_B_* is the total blood flow and *C_v_* is the EV concentration in the venous blood.
(3)$$V_{v}\times \frac{dC_{v}}{dt}=-\left(Q_{B}\times C_{v}\right)+\sum_{i=1}^{n}Q_{i}\times C_{i}/R_{i}$$For venous blood: *V_v_* is the venous blood volume and rest parameter are defined in Equations (1) and (2).
(4)$$V_{a}\times \frac{dC_{a}}{dt}=\left(Q_{B}\times \frac{C_{lung}}{R_{lung}}\right)-\sum_{i=1}^{n}Q_{i}\times C_{a}$$For arterial blood: *V_a_* is the arterial blood volume and rest parameter are defined in Equations (1) and (2).

## 12. PBPK Modeling Application for EV Therapeutics

### 12.1. End and Complicated Life Stage Prediction

PBPK modeling can be used for end and complex life stage prediction in various applications, such as drug development and personalized medicine [97,99]. In drug development, PBPK modeling can be used to predict the pharmacokinetics of drugs in different age groups, including pediatric and elderly populations [127,128]. PBPK modeling can also be used to predict the pharmacokinetics of drugs in special populations, such as pregnant women [129] and individuals with liver [130] or kidney dysfunction [131].

In the context of EV therapeutics, PBPK modeling can be used to predict the end and complicated life stages of EV in the body. This includes the uptake, distribution, metabolism, and excretion of EV-based drugs [93]. The PBPK model can be used to simulate different physiological scenarios, such as variations in body weight, age, and disease state, which can affect the pharmacokinetics of EV-based drugs [132]. Exosomes (EV) have gained significant interest in the field of personalized medicine due to their potential as a liquid biopsy tool [133]. PBPK modeling can be used to predict the pharmacokinetics of drugs loaded in EV in individual patients based on their unique characteristics, such as genomics, proteomics, and metabolomics [99]. PBPK modeling can be helpful in optimizing dosing strategies and to identify potential adverse drug reactions in individual patients [99]. The PBPK model can predict the interactions between EV-based drugs and other drugs in the body [134]. For example, it can be used to predict the impact of drug–drug interactions on the pharmacokinetics of EV-based drugs. PBPK modeling can also be applied in the context of end-of-life care, where it can be used to predict the pharmacokinetics of drugs in patients with advanced cancer or other terminal illnesses [135]. This information can be used to optimize dosing and minimize toxicity, improve patient quality of life, and to make informed decisions about end-of-life care or even in postmortem cases [136,137].

### 12.2. IVIVE

In vitro–in vivo extrapolation (IVIVE) is a widely used approach in the pharmaceutical industry to predict the pharmacokinetics (PK) of a drug in humans based on in vitro data [138]. IVIVE is particularly important in the development of EV therapeutics as there is still limited knowledge on the PK of these nanoscale particles in the human body. PBPK modeling can be used in the context of IVIVE for EV therapeutics. PBPK models can incorporate the physiologic, demographic, and genetic information of an individual to predict the PK behavior of drugs in humans [138]. This information can be used to improve the accuracy of PK predictions and support the development of personalized medicine approaches for EV therapies [133]. PBPK modeling can be particularly useful for EV therapeutics because EVs can be isolated from patient samples and used as a source of in vitro data for PK predictions [9]. Additionally, the large size and heterogeneous nature of EVs can be accounted for in PBPK models, making them a valuable tool for the development of EV-based therapies.

### 12.3. Cancer Model

PBPK models have become increasingly important in the development of cancer therapies as they help to predict the distribution and elimination of drugs in the human body [135,139]. This information can be used to optimize drug dosing and to identify potential toxicity issues early in the development process. PBPK modeling can also be used in cancer research to simulate the pharmacokinetics of anticancer drugs and their interactions with cancer cells [140]. One of the applications of PBPK modeling in the context of EV therapeutics for cancer is the development of cancer models that can be used to simulate the pharmacokinetics of cancer drugs and EVs in the human body [141]. These models can take into account various physiological and pathological factors, such as cancer progression, co-morbidities, and drug interactions, that can affect the pharmacokinetics of both drugs and EVs. By simulating the pharmacokinetics of drugs and EVs in cancer models, PBPK modeling can provide valuable information on the optimal dosing and administration of cancer therapies, as well as the potential toxicity and efficacy of these treatments [142].

### 12.4. Route to Route and Species to Species Extrapolation

PBPK modeling for EV therapeutics can also be used for route-to-route and species-to-species extrapolation. This involves the prediction of pharmacokinetic (PK) profiles of EV therapeutics from one route of administration to another (e.g., oral to IV) [143] or from one species to another (e.g., mouse to human) [144]. Route-to-route and species-to-species extrapolation is an important aspect of PBPK modeling in the context of therapeutic EVs. In the preclinical phase of drug development, drugs are usually tested in animal models, and the PK data obtained from these animal studies are used to make predictions about human PK [144]. The extrapolation of PK data from one species to another and from one route of administration to another can be challenging, as the physiological processes that govern drug PK can differ between species and between routes of administration. PBPK models can help to address these challenges by considering the species- and route-specific differences in the body’s physiological processes that govern PK drugs. By using PBPK models, it is possible to predict the PK of therapeutic EVs in humans based on the PK data obtained in animal studies, and to explore the impact of species- and route-specific differences on the PK of therapeutic EVs.

## 13. Perspective and Future Direction

The use of PBPK models in EV research is still in its early stages, and there is a need for more data and studies to further validate and refine the models. However, PBPK modeling has the potential to play a significant role in the development of EV-based therapies by providing valuable insights into the pharmacokinetics and pharmacodynamics of EVs, which can help guide the design of clinical trials and support the regulatory approval process. While preparing this manuscript, we have attempted to include all relevant studies, but we would like to apologize to any authors whose work may have been inadvertently omitted.

## 14. Conclusions

In conclusion, PBPK modeling is a promising tool for the study of EVs and their therapeutic potential, and its application in this field is expected to grow in the future as the need for a better understanding of the pharmacokinetics and pharmacodynamics of EVs increases.

## Figures and Tables

**Figure 1 biology-12-01178-f001:**
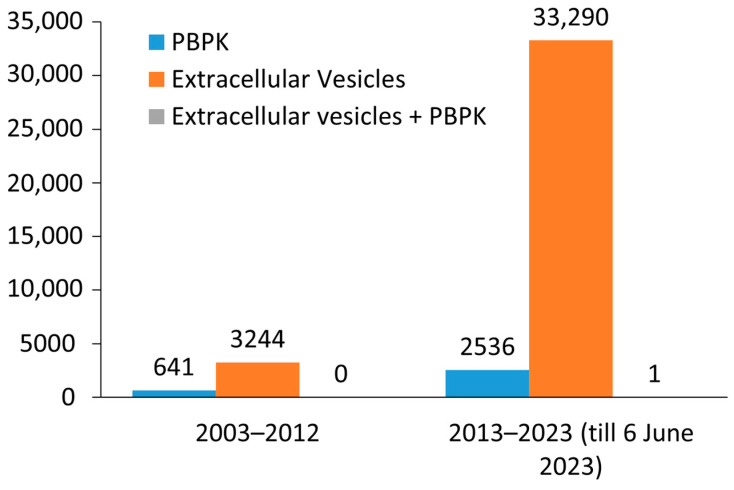
Distribution histogram of the number of scientific literatures on PBPK, Extracellular vesicles and Extracellular vesicles + PBPK retrieved through PubMed on 6 June 2023.

**Figure 2 biology-12-01178-f002:**
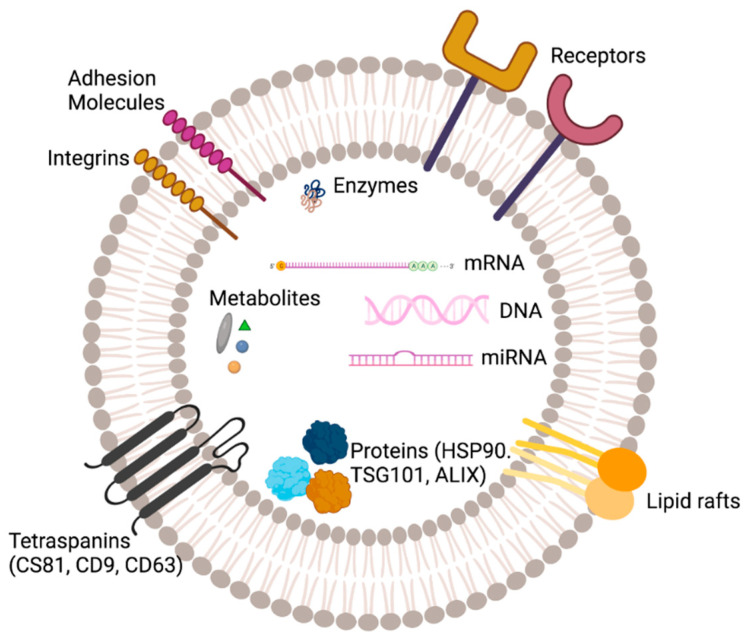
Structure and composition of a generic extracellular vesicle (EV). An EV lumen is composed of proteins, nucleic acid (DNA, mRNA, miRNA), lipid, metabolites and surrounded by a phospholipid bilayer.

**Figure 3 biology-12-01178-f003:**
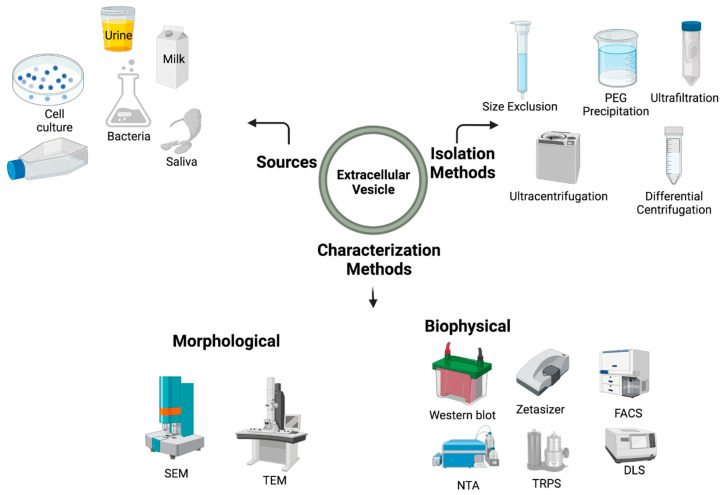
Illustration shows the different sources of EV and technology enhancement in the isolation methods and its characterization.

**Figure 4 biology-12-01178-f004:**
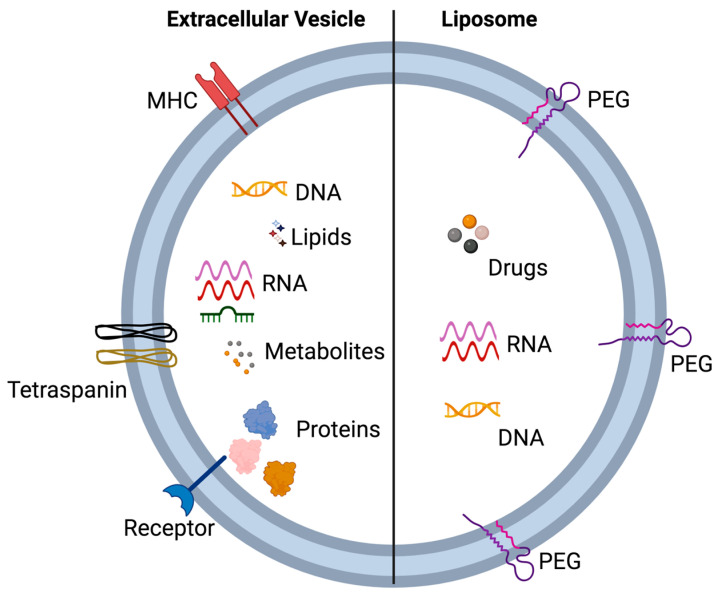
A structural comparison between Extracellular vesicles and Liposomes.

**Figure 5 biology-12-01178-f005:**
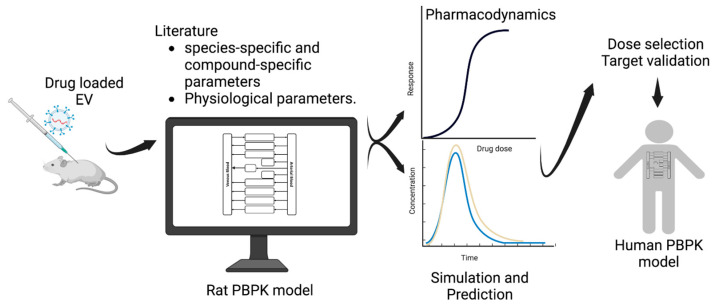
PBPK modeling workflow highlighting its application in extrapolating experimental pharmacokinetic results from mice when injected with drug-loaded EV and then translated to humans [108].

**Figure 6 biology-12-01178-f006:**
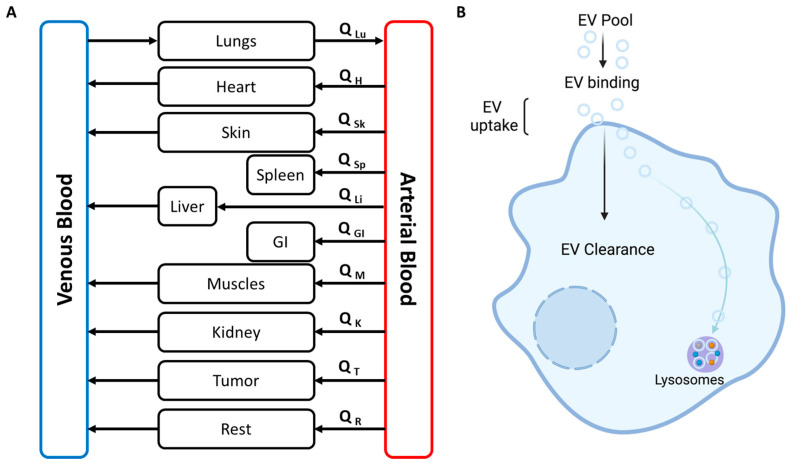
A generic PBPK model (**A**). Mathematical modeling of EV-cell interaction.: illustration shows the interaction of fluorescently labeled EV with the renal cell in Autosomal dominant polycystic kidney disease (ADPKD) [29] (**B**). Note—Q_H_: Venous blood flow in heart. And similar nomenclature is followed for other tissues too.

## Data Availability

Not applicable.

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
