# Peer review of "Physiologically Based Pharmacokinetic Modeling of Extracellular Vesicles"

_biology, 2023, doi:10.3390/biology12091178_

Round 1

Reviewer 1 Report

In this study, Prashant Kumar, Darshan Mehta and John J. Bissler discuss the suitability of the PBPK modeling for its potential application on administration of sEV - exosomes.

Beyond its main topic, the authors describe at the beginning a resumed overview of main techniques related to the study and the biology of sEV (composition, biogenesis, isolation techniques, etc).

I found overall the manuscript very well written, I believe that the topic is relevant, that the PBPK modeling basis and its softwares/mathematical models were properly explained even to a non specialist (like me), and that the perspectives for such approach will grow in the near future, thus making this review a very important publication for the filed.

I have no minor nor major comments/changes for the present form.

Reviewer 2 Report

The Review manuscript “Physiologically based pharmacokinetic modeling of Extracellular Vesicles” has been written well and moderate English language.

Author should include flow chat for the modeling process to be followed, make more understandable to readers.

Author must compare the different modeling approaches rather than only giving introduction.

Author also provides information of type of data needed to make successful modeling model.

Also, Author must add PBPK model validation criteria.

Reviewer 3 Report

In the MS by Kumar P. et al., the authors made a literature review on the Physiologically based pharmacokinetic modeling of Extracellular Vesicles (EVs). In the first part of the MS the authors summarized some of the main features of EVs (i.e., structure, composition, biogenesis) and the current techniques to isolate and to characterize them. After a brief comparison between EVs and liposomes, the authors approached the Physiologically based pharmacokinetic (PBPK) modeling and the main aspects connected to PBPK (i.e., Absorption, Distribution and Clearance), finally focusing on the use of PBPK modelling in the field of drug modified EVs.

The Review is well conceived and written ad it covers an important topic in the field on the EVs, especially in a frontier topic as the use of EVs as a drug delivery system and, as authors stated in the introduction, pharmacokinetics studies covering this topic are really rare in literature.

Collectively, I consider this review suitable for publication in “Biology” even if some small modification should be applied as follow:

1.       In the Introduction section: authors should also refer to the new nomenclature of EVs as small EVs (s-EVs) and large EVs (l-EVs). I suggest the following change to the text (lines 48-50):“Microvesicles (MVs), now also defined as large EVs (l-EVs) [insert ref PMID 30637094], are derived from the plasma membrane and range in size from 0.1 to 1 micrometer. Exosomes, now also defined as small EVs (s-EVs) [insert ref PMID 30637094], are derived from endosomes and range in size from 30 to 100 nanometers [1].

2.       Already in the Introduction section: authors refer to EVs as MVs and exos only, but actually also apoptotic bodies and big EVs derived from cancer cells (i.e., oncosomes) should be reported. I suggest the authors to add the following sentence after the sentence previously mentioned: “In addition, bigger than 1 micrometer EVs were released following apoptosis (i.e., Apoptotic Bodies) or by cancer cells (namely, oncosomes) [add refs PMID 33134295 and PMID 31937439].”

3.       Lines 125-126: intraluminal vesicle is actually referred to the vesicle generated by invagination of endosome membrane leading to the generation of Multivesicular bodies, then the definition given by the authors is incorrect. I suggest resentencing as following:” EVs are composed of a lipid bilayer enclosing an inner lumen encasing bioactive molecules derived from donor cell.”

4.       Sentence corresponding to lines 139-141 is redundant and must be erased.

5.       Lines 161-163: Exos are not generated from the inward budding of plasma membrane but, as authors correctly stated in another point of the MS, from the inward budding of endosome membrane leading to the generation of multivesicular Bodies (MVBs) containing intraluminal vesicles (ILVs). Once MVBs fuse with plasma membrane ILVs were released in the extracellular space as exos. Change the relevant sentence according to.

6.       Line 326: change Evs to EVs.

7.       Line 344: “such as DNA, miRNA, lipids, and proteins.” Is redundant and must be erased.

8.       Lines 487-490: According to point 1, this sentence might be changed in:” S-EVs are able to pass through the endothelial barrier and enter the circulatory system, allowing for distribution to various organs and tissues [87, 88]. L-EVs may be restricted to the micro-vasculature of specific organs and tissues [89].”

9.       Lines 510-512 and 538-39: These sentences are redundant and must be erased.

10.   Line 723: Check for an extra space at the beginning of the sentence:” This Platform….”

Reviewer 4 Report

This is an interesting manuscript, reflecting the good work of the authors in this field of knowledge.

I think this MS can be published after minor corrections:

The Fig.1 can be improved and made in color.

Fig.4 DNA molecules should have the same color. 
